# Freedom to Move: Free Lactation Pens Improve Sow Welfare

**DOI:** 10.3390/ani12141762

**Published:** 2022-07-09

**Authors:** Orla Kinane, Fidelma Butler, Keelin O’Driscoll

**Affiliations:** 1Pig Development Department, Teagasc Animal & Grassland Research and Innovation Centre, Moorepark, P61 C996 Fermoy, Ireland; keelin.odriscoll@teagasc.ie; 2School of Biological, Earth and Environmental Sciences, University College Cork, T12 K8AF Cork, Ireland; f.butler@ucc.ie

**Keywords:** animal welfare, pig production, farrowing crate, free lactation, animal behaviour

## Abstract

**Simple Summary:**

Pigs in Ireland are reared in intensive indoor systems, with sows normally confined in farrowing crates for a period of five weeks each time they farrow. This practice presents major animal welfare problems, with permeant confinement being banned at all other stages of production due to the negative impact on welfare. This study identified positive effects on sow behaviour and welfare when housed in free lactation pens compared with conventional farrowing crates. Such pens allow sows greater freedom of movement throughout farrowing and lactation, and this resulted in decreased locomotion score impact for sows housed in the free lactation treatment. This could be beneficial with regard to both sow longevity and sow welfare.

**Abstract:**

Farrowing crates present a major animal welfare problem. This study investigated the effects of temporary confinement at farrowing on sow welfare and aimed to determine whether this type of system could improve sow welfare through increased freedom of movement. Sows were housed in one of two farrowing accommodation treatments: conventional farrowing crates (Control) or free lactation pens (Free). Sows in the Control treatment were confined from entry to weaning, a period of five weeks. Sows in the Free treatment were temporarily confined from before farrowing (approximately 24 h) until day 4 post-partum, after which time the crate was opened, and they had increased freedom of movement. Sow physical measures (weight, back-fat thickness, hoof score, locomotion score and tear stain score) were measured at entry to farrowing accommodation and at weaning. Salivary cortisol concentration was measured throughout lactation. Farrowing duration and sow posture (Days 1, 3, 7 and 34 after entry) were recorded. Between entry and weaning, locomotion scores significantly increased for sows housed in the Control treatment compared with those housed in Free lactation pens (*p* < 0.01). Sows in the Free treatment were observed to use all orientations in the pen, showing that when more space is made available to them, they will choose to utilise the space. Tear staining under the left eye was found to be less in Free sows at weaning (*p* = 0.05), indicating reduced stress. However, salivary cortisol concentration was higher in Free sows overall; cortisol is affected by both positive and negative stimuli, and so, this may be due to factors other than stress, such as higher levels of activity and mental stimulation. These results suggest that free lactation pens can benefit sow welfare; increased freedom of movement throughout lactation can improve sow locomotory health, and as suggested by improved tear stain scores, sow stress levels may be reduced in this type of system compared with conventional farrowing crates.

## 1. Introduction

The use of farrowing crates in pig farming is increasingly topical, as the general public become more aware of animal welfare issues on farms. Permanent crating of sows at the time of farrowing and lactation has led to public concern with regard to sow welfare [1,2]. The European Citizens’ Initiative ‘End the Cage Age’ collected 1.4 million signatures from supporters in 28 member states in 2021 and called on the European Commission to propose legislation to prohibit the use of farrowing crates [3]. Indeed, 94% of Europeans believe it is important to protect the welfare of farmed animals [4]. It is scientifically recognised that farrowing crates negatively affect sow welfare [5], as they prohibit locomotion completely and can have a negative impact on physical comfort [6], as evidenced by the high prevalence of shoulder sores in restricted sows [7]. Moreover, this type of system can also induce mental stress, as it prevents sows from performing their normal behaviour. Specifically, farrowing crates prevent direct social contact with other sows and the interaction between the sow and piglets, the choosing of a nest site, the opportunity to perform rooting and nest building behaviour, isolation during farrowing, the possibility for exploration and the choice to defecate away from the resting area [8,9,10,11,12].

However, farrowing crates are attractive to producers, as they can protect piglets from crushing, and they ensure the use of as little space as possible. They also enable quick, safe and easy checking of the animals by the stockperson. The improved survival of piglets is the major reason for their use, and indeed, a meta-analysis found a 14% increase in relative risk of piglet mortality in farrowing pens compared with crates [13]. However, although mortality due to crushing is generally found to be higher in loose farrowing systems than in crates [14], other causes of mortality can be higher in farrowing crates than loose systems, and some studies have found equal piglet survival rates overall (e.g., 1.40 piglets/litter in free farrowing pens vs. 1.42 piglets/litter in crates [15]). Indeed, the authors of the current study found that pre-weaning percentage mortality was similar in free lactation pens (≈16.0%) and standard crates (≈14.5%) [16]. However, the causes of death differed, with more crushing in the free lactation pens but more deaths due to illness and culling in the standard pens.

Protection from crushing is ever more important with the increases in litter sizes that have occurred due to selection for higher sow productivity [17,18]. The ability of sows to farrow and nurse large litters is essential in modern pig production [6]. However, larger litters are associated with lower birthweights and a greater proportion of vulnerable piglets that are at high risk of crushing, particularly if there is less space in the pen per piglet [19]. Since the first three days post farrowing are the most critical period for crushing [20], restricting the sows’ movement during this time may be a solution.

Animal welfare science has shown that confinement can lead to severe stress in sows [21], and this may also have negative implications for production. Sows housed in farrowing crates are unable to perform their natural farrowing and maternal behaviours [22], which can cause frustration and lead to increased farrowing durations [23]. These animal welfare concerns along with social pressure [24] have resulted in increasing interest in the development of alternative farrowing systems [25]. Although considerable research is on-going on free farrowing systems (Google scholar search identified 4490 papers published since 2018), the UK Farm Animal Welfare Council [26] concluded that satisfactory results are not yet available, and commercial developments are not yet sufficiently advanced to allow recommendation of compulsory replacement of farrowing crates. Hansen [27] tested ten different designs of farrowing pens for loose-housed sows and recommended there are still challenges to be resolved before implementing this type of management system on a broad scale.

Pig producers are more likely to consider implementing a system that can deliver acceptable levels of piglet mortality [11]. Temporary confinement around the day of farrowing allows for more controlled management of sows than fully loose housing while providing the same level of protection for piglets at their most vulnerable. This system of management may be a compromise, which can ensure current production levels are maintained while also improving sow welfare. The lifetime performance of commercial sows relies on longevity, which is dependent on good health. Locomotory issues account for 13% of all sow cullings, and over half of these females have not yet attained their second parity [28]. Therefore, employing a system of farrowing and lactation management, which does not exacerbate locomotory problems is important for both the welfare and the productivity of sows.

This study aimed to determine whether sow welfare could be improved using free lactation crates compared to conventional crates. Although there has been much work recently on the benefits of loose housing, uptake by producers has not followed due to concerns around piglet mortality. This study investigated a system, which allows for temporary confinement, meaning both the welfare needs of the sow and the piglets are catered for. We hypothesised that a temporary crated system would provide benefits to sow welfare, while minimising piglet losses, and thus be a more manageable change for producers as the industry transitions towards loose housing.

## 2. Materials and Methods

This study was carried out in the Teagasc Moorepark Pig Development Research Facility, Co. Cork, Ireland. The study was approved by the Teagasc Animal Ethics Committee (TAEC192-2018). The research farm has high herd health status, vaccinates against clostridia, erysipelas, parvovirus and mycoplasma, and is free of all other disease.

### 2.1. Treatments and Experimental Design

Four farrowing batches (26–30 sows/batch) were used in the experiment. From each batch, 12 sows (Large White × Landrace) in good general health and showing no signs of clinical lameness were selected for the study (*n* = 48 sows in total), at day 108 of gestation (day-1 of the experiment; D-1). This was the day prior to movement from gestation housing to the farrowing rooms (D0). Gestating sows were managed in a dynamic group pen, which held 120 animals at any one time. The pen had two electronic sow feeders (ESF; Schauer Feeding System (Competent 6), Prambachkirchen, Austria), insulated concrete lying bays and fully slatted floors. Water was available to sows ad libitum from single-bite drinkers in the ESF’s and from five drinker bowls located around the group pen. Within each batch, sows were assigned to one of six blocks on the basis of locomotion score (1.5 ± 0.51 (1–2)) (using the system described in Hartnett et al. [28] ranging from 0 (perfect) to 5 (unable to move), parity (2.57 ± 2.01 (1–6)), teat number (15.15 ± 1.15 (14–18)), weight (275.69 ± 39.85 (188–358)) and back-fat thickness (17.02 ± 3.63 (10–26)) (Back-fat thickness was measured using a digital back-fat indicator (Renco Lean-Meater, Renco Corporation, Golden Valley, Minneapolis). Two points 6.5 cm from the central dorsal line and in line with the last rib were shaved, the back-fat measured, and the average of the two measurements was recorded). One sow from each block was then randomly assigned to one of two treatments: Control or Free (i.e., six sows per treatment per batch).

Treatment pens were in one of three farrowing rooms. One room contained six Free pens. Two other rooms contained seven Control pens. Within each batch, only one of the Control rooms was used, and only the 6 Control sows within that batch were housed in the room (i.e., the 7th farrowing pen was left empty).

The Control treatment consisted of conventional farrowing crates, which were installed in farrowing pens measuring 184 × 250 cm (4.6 m^2^) (Figure 1). The Free treatment consisted of a similar crate, in a slightly larger pen (212 × 261 cm, 5.5 m^2^). In the Control treatment, the crate confined the sow and allowed for extremely minimal movement, allowing the sow to stand and lie but not to turn or move around the pen. The crate in the Free pens allowed for the sow to be confined as before; yet, the crates could also be opened to allow the sow increased freedom of movement. When the crate was opened, the sow could freely turn around through 360°.

Farrowing rooms were artificially lit from 07.00 to 16:30. Sows were fed and feed intake recorded using a computerised feed delivery system (DryExact Pro, Big Dutchman, Vechta, Germany). Sows were fed twice daily from D1 (entry to the farrowing room) to day 14 of lactation and three times daily thereafter until weaning. The sow lactation feeding curve started at 2.9 kg/d at day 0 of lactation and gradually increased to 6.3, 7.8, 8.7 and 8.2 kg/d, on average, at days 7, 14, 21 and 26 of lactation, respectively. Feed troughs were checked once per day in the morning to assess sow feed intake, and individual feeding curves were adjusted daily by increasing or decreasing the feed allowance by 5% depending upon whether there was feed wastage or the trough was completely emptied. Water was provided on an ad libitum basis to sows from a single-bite drinker in the feed trough and to suckling piglets from a bowl in the farrowing pen. Farm staff were present on the farm from 07.00 to 16.30 each day to assist with farrowing and provide general care to the animals. One sow was removed from the trial due to a shoulder lesion. Twelve piglets were removed from the trial due to health and welfare issues, such as hunger or injury, assessed as part of routine management by experienced farm staff. These piglets were moved to a nurse sow and were not reintroduced.

### 2.2. Animals and Management

Sows were introduced to the farrowing rooms on D1. Sows in the Control treatment were confined in the crate from entry until weaning, a period of five weeks. In the Free treatment, the farrowing crates were initially left open, so that sows were loose and able to turn around in the pens. From the afternoon of D5 (16:00), the crates were closed to confine the sows overnight and to allow for habituation to the crate, then opened again each morning (08:00). When sows in the Free crates were observed to be producing milk (an indication that they were close to farrowing), the crate remained closed. Free sows remained confined from the first sign of milk until the morning of day 4 post farrowing (08:00). After this period, the crate remained open, so sows were allowed freedom of movement until weaning. Farrowing was not induced.

Piglets were ear tagged at birth to allow for identification. Sex and birth weight were recorded within the first 24 h. Cross fostering was carried out where necessary to ensure that there was never a greater number of piglets than functional teats. This took place within the first 48 h, and the identities of both the birth and foster sow were recorded. Records of mortality and its cause were kept and updated daily. If crushing of a piglet was observed, an intervention to save the piglet was always attempted (i.e., attempt to move the sow to release the piglet), as is normal farm practice.

### 2.3. Physical Measures

#### 2.3.1. Body Weight and Back-Fat Thickness

Body weight and back-fat were recorded on D0 (the day prior to entry to farrowing rooms) and again on the day of weaning, D35 of the experiment (26.5 ± 1 days post farrowing). Each sow was weighed using an electronic sow scales (EziWeigh 7i, O’Donovan Engineering, Co. Cork, Ireland). To calculate empty weight prior to farrowing, the following equation was used:$$empty\;farrowing\;weight=\left[weight\;at\;d108-\left(total\;born\times 2.25\right)\right]$$

The value of 2.25 kg is an estimate of the increased weight in the gravid uterus and in mammary tissue, attributed to each pig in a litter [29]. Back-fat thickness was measured using a digital back-fat indicator (Renco Lean-Meater, Renco Corporation, Golden Valley, Minneapolis). Two points 6.5 cm from the central dorsal line and in line with the last rib were shaved, the back-fat measured, and the average of the two measurements was recorded.

#### 2.3.2. Locomotion Score

Sows were locomotion scored on D0 and on the day of weaning, D35. Locomotory ability was scored while the animals walked on a solid concrete corridor for a distance of at least 10 m, from the front, rear and side of the animal. All observations were carried out by one trained observer, using the system described in Hartnett et al. [28] and ranged from 0 (perfect) to 5 (unable to move).

#### 2.3.3. Hoof Score

Hoof score was recorded for all sows on D1 (i.e., the first day in the farrowing rooms) and on the day of weaning, D35. Hind hooves were scored. Scoring on D1 was carried out when sows were lying down, and hooves were visible to the observer. At weaning, hoof scores were recorded by raising the sows 0.75 m above the ground using a hydraulic chute (FeetFirst Sow Chute, Zinpro Performance Minerals, Eden Prairie, MN, USA). The medial and lateral toes, medial and lateral dew claws, sole and heels of both hind feet were inspected, and the severity of the following lesions was scored: heel overgrowth and erosion, heel–sole crack, white line damage, dew claw length, dew claw cracks, vertical cracks, horizontal cracks and toe length. The scoring system was a modified version of the FeetFirst claw lesion scoring guide from Zinpro Corporation, described in Hartnett et al. [28].

#### 2.3.4. Tear Stain Score

Sow tear stain scores were recorded on D1 and on the day of weaning, according to the DeBoer–Marchant–Forde scale described in Deboer et al. [30]. Excess dirt was initially removed from the eye area using warm water to provide for a standardised baseline as much as possible and thus allow for more accurate measurement of staining throughout the time spent in the farrowing pens. Each eye was scored separately according to the following scoring system: 0 = no visible stains, 1 = barely detectable stains not extending below eyelid, 2 = visible stain < 50% in ratio to the eye, 3 = visible stain 50–100% in ratio to the eyes, 4 = visible stain > 100% in ratio to the eye but not extending below the mouth line, 5 = visible stain extending below the mouth line [30].

#### 2.3.5. Salivary Cortisol

One saliva sample was collected from each sow between 09:00 and 10:00 on each of ten collection days. The first was collected on D1, when sows were waiting in the collection area outside the group gestation pen. Subsequent samples were collected on days 2, 3, 5 and 6 after entry to the farrowing rooms, on days 5, 7, 14 and 21 after farrowing, and on the day of weaning. On days 2, 3 and 5, sows in the Free treatment were not confined in the crates. On the morning of day 6, they had been confined overnight for the first time. On day 5 post farrowing, the crate had been opened overnight for the first time since farrowing and remained so for the rest of lactation. Saliva samples were collected by allowing the sow to chew on a cotton bud (Salivette, Sarstedt, Wexford, Ireland) for 30 to 60 s until it was thoroughly moistened. Samples were placed in plastic tubes, stored at 8 °C for no longer than 5 h, then centrifuged at 1500 rpm for 25 min and stored at −20 °C until analysis.

At analysis, samples were defrosted, centrifuged and analysed in duplicate using a commercially available salivary cortisol assay kit (Expanded range high sensitivity salivary cortisol enzyme immunoassay kit, Salimetrics Europe Ltd., Suffolk, UK), according to the manufacturer’s procedure and as per previous studies investigating salivary cortisol in pigs [31,32,33,34,35]. Samples were randomly spread across each plate, so that there were 19 Control and 19 Free samples. Cortisol was detected at a minimum concentration of <0.003 µg/dL. The inter-assay CV (*n* = 16 plates) was 32%, and the intra-assay CV (*n* = 443 samples) was 8.83%.

#### 2.3.6. Farrowing Duration

All farrowing pens were recorded continuously by video cameras (QVIS HDAP400 CCTV cameras and a Pioneer-16 digital recorder case, CCTV Ireland, Kildare, Ireland) from entry to farrowing room until all sows had finished farrowing and on D34 of the experiment (day 26.5 ± 1 of lactation). Farrowing duration was extracted from the videos by observing each sow continuously from birth of the first piglet until birth of the last. From this, the total farrowing duration and the interval between the birth of each individual piglet were recorded.

#### 2.3.7. Sow Posture and Orientation

Video footage from D1, D3, D7 and D34 (day 26.5 ± 1 of lactation) was observed. On D1 and D3, sows in the Free treatment were loose in the pen since entry. On D7, sows had been confined overnight for 3 nights. Videos were observed using scan sampling at 5 min intervals between 11:00 and 17:00 (73 samples/sow/day). At each time point, the posture of the sow as well as her orientation were recorded. Postures were as follows: stand, sit, lie on the belly, lie on the left side, lie on the right side. Orientation was considered 1 to 12 as per the position of numbers on an analogue clock face; position 12 was oriented with the head directly facing the feeder. This was recorded for Free sows only because sows in the Control treatment were always oriented towards the feeder. The times spent lying on the left and right sides were summed to provide a total figure for lying on the side. The percentages of observations that sows spent in each posture and orientation were then calculated across all the observations within each recording day.

#### 2.3.8. Proximity of Piglets and Response to Separation from Piglets

The percentage of piglets in contact with the sow was also recorded at 5 min intervals between 10:00 and 16:55 inclusive on D1, D3, D7 and D34. The responsiveness of the sow to her piglets was estimated by carrying out a separation and return test on day 21–22 of lactation. Piglets were removed from the pen for 2 h to ensure that they had missed approximately 2 nursing bouts. All sows were encouraged into a standing position immediately prior to the piglets being returned to the pen. The time that it took the sow to lie down and then to nurse the piglets was recorded.

### 2.4. Statistical Analysis

Statistical analysis was carried out using SAS (v 9.4, SAS Institute Inc., ((Cary, NC, USA), 1989), and the sow was considered the experimental unit. All data were tested for normality prior to analysis by examination of histograms and normal distribution plots using the univariate procedure. When linear models were used, residuals were inspected after analysis to confirm normality. Model fit was determined by choosing models with the minimum finite-sample corrected Akaike information criteria (AIC). Degrees of freedom were estimated using the Kenwood–Rogers adjustment. Results were deemed statistically significant when α level was below 0.05, and a tendency was considered when α level was between 0.05 and 0.1. Either the Tukey–Kramer or Bonferroni adjustments were used for multiple comparisons where least squares means (LS means) were determined and *p*-values were adjusted. Data are presented as LS means and standard errors.

#### 2.4.1. Physical Measures

Sow weight, back-fat and feed intake during lactation were analysed using general linear models, with treatment, sow parity and replicate included as fixed effects. Parity was classified as either first parity or greater than first parity in all models. Empty weight at farrowing was considered a covariate for weaning weight, and for the other measures, values recorded at assignment to treatment were considered covariates. Feed intake was compared based on total intake and average intake per day.

The Mann–Whitney test (Proc Npar1Way) was used to compare locomotion scores at weaning, as on entry to the farrowing rooms, half of the animals in each treatment had a score of 1, and half had a score of 2. The total hoof score (i.e., sum of the individual measures for all four claws) was analysed using a general linear model. Fixed effects were as before, with the addition of inspection (i.e., entry to the farrowing crate and weaning) and the interaction between inspection and treatment. Inspection was considered a repeated effect, and a compound symmetry covariance structure was specified. For analysis of the individual hoof scores, a generalised linear model was used (Proc Genmod), with the same fixed and repeated effects as before. A multinomial distribution was specified, with a cumulative logistic link statement.

The Mann–Whitney test (Proc Npar1Way) was used to compare tear stains for both the left and right eyes at entry to the farrowing rooms and at weaning. Left and right eyes were analysed separately, as previous studies have shown differences in tear staining for both eyes in response to stressors [30].

Salivary cortisol was measured using a general linear model, with the same fixed effects as before (treatment, parity, collection day and replicate). The initial sample was used as a covariate. Collection day was considered a repeated effect, with an autoregressive covariance structure. The EIA plate was included as a random effect. Due to the extremely large number of multiple comparisons, a post hoc Bonferroni test for multiple comparisons was applied to only the raw *p*-values calculated between treatments on each sampling day.

#### 2.4.2. Behaviour

Farrowing duration and the interval between birth time of piglets were both analysed using a general linear model, with the same fixed effects as before. For the birth interval, the birth order of the piglet was also included, and this was also considered a repeated effect, with an autoregressive covariance structure. Total number born was also included as a covariate, as this could not be controlled for in the experimental design. The interval between piglets was log transformed, so that residuals approached normality.

The percentage of observations that sows spent in each posture was analysed using a general linear model. Recording day was included as a repeated measure with an autoregressive covariance structure. The percentage of piglets that were in contact with the sow was analysed using a similar model but without the repeated effect of day.

The percentage of recordings that sows spent with their head pointing towards each direction was analysed separately using a general linear model as before. Both direction and day were considered repeated effects, and as such, a direct product autoregressive correlation structure was used. The number of transitions between positions per hour was also calculated and analysed using a similar model, without repeated effects.

The time to lie and time to nurse piglets after the separation and return test were analysed using a general linear model. Only data from the first three replicates were available for this analysis.

## 3. Results

### 3.1. Physical Measures

#### 3.1.1. Weight, Back-Fat Thickness and Feed Intake

There was no effect of treatment on any aspect of live weight, back-fat depth measurement or feed intake (Table 1).

#### 3.1.2. Locomotion Score

At entry to the farrowing rooms, 50% of sows in both treatments had a score of 1, and 50% had a score of 2. Both the locomotion score at weaning and the difference in locomotion score between entry and weaning were affected by treatment (*p* < 0.01 for both), with scores being higher, indicating more impaired locomotion for Control sows than Free. The percentages of sows in each treatment that had a score of 1, 2, 3 and 4 at weaning are shown in Figure 2.

#### 3.1.3. Hoof Score

Treatment had no effect on total hoof score (i.e., the sum of the individual scores for each disorder; *p* = 0.69), but there was an effect of inspection (*p* < 0.001), with sows having higher (worse) scores at exit (41.57 ± 1.19) than when they entered the farrowing room (36.29 ± 1.19). The difference tended towards significance for sows in the Control treatment (*p* = 0.07) and was significant for Free treatment sows (*p* < 0.01). However, there was no interaction between the examination time (entry and exit to the farrowing room) and treatment (*p* = 0.43). There was also an effect of parity (*p* < 0.05), with the hoof score of sows that were farrowing for the first time being lower (i.e., better; 36.52 ± 1.94) than sows from all other parities (41.34 ± 0.93).

With regard to the individual disorders, which were investigated (heel overgrowth and erosion, heel–sole crack, white line damage, dew claw length, dew claw cracks, vertical cracks, horizontal cracks and toe length), there was no effect of treatment on any of the disorders or interaction between treatment and inspection time. There was, however, a tendency for higher heel erosion and heel–sole crack scores in Free sows compared with Control (Table 2).

#### 3.1.4. Tear Stain Score

At entry to the farrowing rooms, there was no difference in tear stain scores between treatments for either the left eye or the right eye (Table 3). However, by the end of the experiment, although there was no effect of treatment on tear stain score for the right eye, sows in the Free treatment had lower tear stain scores around the left eye than those in the Control.

#### 3.1.5. Salivary Cortisol

Salivary cortisol tended to be higher in Free sows (0.341 ± 0.023 μg/dL) than Control (0.279 ± 0.023 μg/dL; *p* = 0.062). There also tended to be an interaction between treatment and sampling day (*p* = 0.09; Figure 3. On the second day after entry to the farrowing pens, sows in the Free treatment had higher cortisol levels than Control (*p* < 0.05), and they tended to have higher levels on the day after the crates were opened post farrowing (*p* = 0.09).

### 3.2. Behaviour

#### 3.2.1. Farrowing Duration

There was no effect of treatment on farrowing duration; in total, Free sows took 07:43:49 ± 01:16:55 to farrow, whereas Control sows took 07:45:42 ± 01:15:25 to farrow. Neither was there a difference in farrowing interval (Free = 00:07:14, Control = 00:08:47, (back-transformed least-squares means)).

#### 3.2.2. Sow Posture

There was no effect of treatment on the proportion of time that sows spent standing (*p* = 0.70) or sitting (*p* = 0.45). Overall, Free sows tended to spend more time lying on their bellies (*p* = 0.07), and Control sows spent more time lying laterally (*p* < 0.05). The amount of time spent lying laterally was also investigated as a percentage of the total time spent lying. Here, again, sows in the Control treatment spent a higher proportion of lying time on their side (76.32 ± 0.04%) than sows in the Free pens (65.08 ± 0.04%; *p* < 0.05).

Although there was no interaction between the time spent lying on the belly or laterally and recording day, numerically, the time spent lying on the belly increased across time for sows in the Control treatment but not in the Free treatment, whereas time spent lying laterally decreased for Control sows but not for Free.

It was found that sows in the Free treatment tended to spend less time lying on the left side than sows in the Control treatment (*p* = 0.01). Although there was no interaction between treatment and time, it appeared that this was driven by a higher proportion of lying time spent on the left side in Control sows on D1 and D3, which corresponds to the initial time spent crated after entry to the farrowing pens. During these two days, Free sows were not confined in the farrowing crate.

#### 3.2.3. Orientation in the Pen

The orientation of each sow in the Free treatment was recorded as though the sows’ head pointed towards the numbers on a clock face (i.e., 12 representing the sows’ head oriented directly forward in the pen towards the trough and 6 representing the sows’ head pointed towards the back wall of the pen). Overall, sows spent the highest proportion of observations oriented directly towards the front of the pen (40.4 ± 2.2%), indeed, significantly more than in any other orientation (*p* < 0.001 for all comparisons). This was followed by having their head oriented towards position ‘1’ on a clock face, then directly towards the rear of the pen, position ‘6’ (21.2 ± 2.2%, and 15.0 ± 2.2%, respectively).

There was an interaction between day and the orientation of the sows (*p* = 0.001). The percentage time that sows spent oriented towards each number on a clock face, on each day, can be seen in Figure 4. The only significant difference between the proportion of time spent oriented in any direction between days was between the time spent oriented directly towards the feed trough on D3 and D7 (*p* < 0.05) and on D7 and D34 (*p* < 0.001). The highest percentage was on D7, which represents a day prior to farrowing, when sows had been confined in the crates the previous night, and the lowest percentage was on D34, when sows were approximately 3 weeks into lactation. The number of times each sow changed orientation increased as the experiment progressed (*p* < 0.05), and indeed, there was a significant difference in the number of transitions between D1 and D34 (*p* < 0.05).

#### 3.2.4. Proximity of Piglets

The percentage of piglets in contact with the sow, recorded at 5 min intervals between 10:00 and 16:55 inclusive on D1, D3, D7 and D34, was not affected by treatment (Control = 46.5 ± 3.0%, Free = 46.1 ± 3.1%; *p* = 0.88). However, there was an effect of hour of the day (*p* < 0.05) and an interaction between treatment and hour of the day (*p* = 0.01). In general, piglets in the Free treatment were observed in contact with the sow more often in the morning and late afternoon, whereas in the Control treatment, there was a peak in early afternoon. Although there was no significant difference at any time point, more piglets in the Control treatment tended to be in contact with the sow in the hour leading up to 14:00 than in the Free treatment.

#### 3.2.5. Response to Separation from Piglets

There was no effect of treatment on either the time it took sows to lie down after being separated from their piglets or the time it took to nurse them.

## 4. Discussion

Several physical and behavioural measures were recorded over the course of the study period in order to holistically assess sow welfare, with several pointing towards improved welfare for sows in the free lactation pens. Sows housed in free lactation pens had improved locomotory scores at weaning when compared with Control sows. Tear stain scores of the left eye were lower in Free sows at weaning, although salivary cortisol concentrations were higher. Finally, sows in free lactation pens made use of the greater freedom of movement by utilising all orientations in the pen.

The physical condition of the sow is extremely important not just for welfare but also with regard to performance. Sows that lose excessive body condition during lactation have impaired reproductive performance subsequently [36], and thus, the finding that there was no difference between treatments (Table 1) is important from the perspective of economic sustainability of both systems. These measures can also give an indication of general welfare, and thus, again, it is positive to find no difference in weight loss or back-fat thickness loss of the sows in the free lactation pens relative to the standard farrowing crates. This was the case, even though sows in the Free pens appeared to be more active, as they were able to orient around the pen. Thus, any impact of increased activity did not have a negative impact on body condition or result in increased feed intake. Sow body lesions were not recorded, but in a similar study carried out by Ceballos et al. [37], a reduction in teat lesions in sows housed in temporarily confined farrowing accommodation was found, again indicating improved welfare in a system allowing for greater freedom of movement of the sow.

At the same time, there was a negative impact on hoof health when sows were managed in the free lactation pens compared with standard crates. Although hoof damage scores increased in both treatments between entry and exit to the system, the increase was only significant for sows in the free lactation system. Hoof damage in sows is generally a consequence of mechanical damage, and the ability to move more freely and often in the free lactation system could thus have been an underlying cause. The aspects of hoof health, which appeared most affected were erosion to the heel and damage to the joinbetween the heel and the sole, both of which can be associated with wear and tear of the tissue of the foot. Nevertheless, although the level of hoof damage was higher in sows in the free lactation pens, sows in this system had better locomotory ability at weaning than sows in the standard crates. This could indicate that maintaining some level of movement throughout lactation prevents sows from developing stiffness in their limbs.

On exit from the farrowing rooms, sows from the free lactation accommodation had lower tear stain scores for the left eye than those housed in conventional farrowing crates. Tear stain score (chromodacryorrhea) is a measure of stress commonly used in laboratory rats and more recently in pigs [38]. Telkänranta et al. [39] found tear staining to correlate with ear and tail damage. Deboer et al. [30] found that isolation and lack of enrichment resulted in higher tear stain scores, and Chou et al. [40] also found a correlation between tail damage and tear stain scores. It is therefore possible that the sows, which were housed in free lactation pens and showed lower levels of tear staining of the left eye at weaning, experienced less stress throughout the period of farrowing and lactation than those housed in conventional farrowing crates.

Although it was anticipated that sows in the free lactation treatment would experience less stress, and thus have lower salivary cortisol levels than those in the standard farrowing crates, the opposite was found; overall, there was a tendency for higher cortisol levels in sows housed in the free lactation pens than those housed in conventional farrowing crates. Indeed, this was the case particularly on days when the opposite response was expected—on the second day after movement to the farrowing pens (i.e., sows in the conventional system had at this point been confined for 48 h) and on the day after the crate had been opened post farrowing (i.e., a day when the sows in the free lactation system had experienced freedom of movement again for 24 h, after being confined for 3 to 4 days). In a study, which analysed hair cortisol in sows housed in farrowing crates or a loose housing system, Wiechers et al. [22] found no difference and concluded that confining sows in farrowing crates did not affect chronic stress levels. Grimberg-Henrici et al. [41] found higher levels of cortisol in group-housed sows compared with individually crated sows, which might not be expected, as these sows do not experience the effects of confinement and isolation. They proposed increased physical activity as the cause. This helps explain the results of the current study; as evidenced in the behaviour recording, sows housed in the free lactation system utilised the space available to move around, and thus, the opportunity to be more active could have increased salivary cortisol concentrations of Free sows. Ceballos et al. [6] also found that sows utilised the space made available to them in a similar study where a hinged farrowing crate was opened after farrowing.

It could be expected that sows, which are less stressed around the time of farrowing may have shorter farrowing durations. However, contrary to this hypothesis, no effect of treatment with regard to farrowing duration was found. This is, nonetheless, an encouraging result, as it shows that sows housed in free lactation pens are not affected negatively regarding farrowing duration. Research suggests that confined sows exhibit an increased level of cortisol prior to farrowing [42], resulting in an extended farrowing duration. Recent work by Nowland et al. [43], which also used temporary confinement, has resulted in similar findings to the current study, with no effect of treatment on farrowing duration being observed (OPEN, crates were open until the sow stood following parturition; CLOSED, crates were closed throughout parturition). Moreover, the sows in the free lactation pens had higher cortisol levels in the days after the move to the farrowing housing, even though they were not confined. Regardless of the reason for cortisol levels being higher (e.g., whether due to a relatively positive or negative affective state compared to the standard crate treatment), the fact that the cortisol level was higher does suggest that there is some difference in welfare experienced by the sow between treatments.

Moreover, because farrowing is already a stressful event, it is possible that once the sow has begun to farrow, the environment has limited influence over the stress response. Indeed, Ref. [44] found that housing type (farrowing crate, pen with sawdust, pen with abundant nesting material) did not affect oxytocin concentrations in sows during farrowing and, interestingly, found farrowing duration to be shorter in sows with confinement than those not confined.

Although the proportion of time sows spent standing or sitting was not affected by accommodation type, sows housed in free lactation pens tended to lie on their bellies more, with sows in conventional crates tending to spend a greater proportion of time lying laterally. Interestingly, sows in conventional crates spent significantly more time lying on the left side than those housed in free lactation pens. There is no explanation for this difference referred to in the literature, and it may be possible that these sows were orienting towards the window in the farrowing room.

Regardless of whether sitting or lying, sows made use of all possible orientations in the free lactation pens. During most of the observations, they were recorded as facing either the front or the back of the pen. This was most likely due to having most space along this line, as even though they could turn around completely, the width of the crate may not have been sufficient for them to lie comfortably across even when open. Thus, when given the opportunity, sows will remain more active during lactation than is possible in conventional farrowing crates. Indeed, the number of transitions per hour increased as lactation progressed. Reduced space allowance triggers stress responses in farmed animals [45], and increased space allowance results in a reduction in damaging behaviour in pigs [46]. The option to express a wider range of natural behaviours is generally accepted to improve welfare [47], and so, being given the option to express some of their natural locomotory behaviour at this time is likely positive with regard to sow welfare.

The percentage of piglets in contact with the sow was not affected by treatment. In another study, Loftus et al. [5] observed Free farrowing sows to spend more time nursing their piglets, socialising with their piglets and exploring the pen. It should be noted that in the current study, sows in the free lactation treatment had the opportunity to move away from or push away piglets, but they spent the same amount of time in contact with their offspring as those housed in farrowing crates. This could be indicative of a higher level of maternal care in the Free sows.

## 5. Conclusions

The use of free lactation pens improved sow welfare through lower (better) locomotion score at weaning when compared with sows housed in conventional farrowing crates. This is an important finding, with implications for longevity and production and for sow physical comfort. Furthermore, a greater range of expressions of natural locomotory behaviour was observed in free lactation sows, making use of the space available to them and occupying all orientations in the pen. This demonstrates that sows, when allowed to do so, will remain more active around farrowing and lactation than they are capable of currently in extremely restrictive farrowing crates. Lower levels of left eye tear staining seen in free lactation sows at weaning suggested reduced levels of stress. Salivary cortisol was higher in Free sows. However, it is important to note that cortisol is an indicator of arousal rather than stress and can be heightened as a result of both positive and negative stimuli, for example, increased activity and increased interaction with piglets for Free sows.

The use of free lactation crates can be seen from this study to improve sow welfare in some regards. Stress levels as assessed using tear stain scores and salivary cortisol concentrations must be interpreted carefully. It is possible that the sows in free lactation accommodation were not in fact experiencing greater levels of negative stress but were simply more active than their counterparts housed in farrowing crates. The sows in the free lactation pens had improved ability to interact with their piglets, and this may have resulted in greater mental stimulation when compared with confined sows in farrowing crates that could not turn around to interact with their piglets.

## Figures and Tables

**Figure 1 animals-12-01762-f001:**
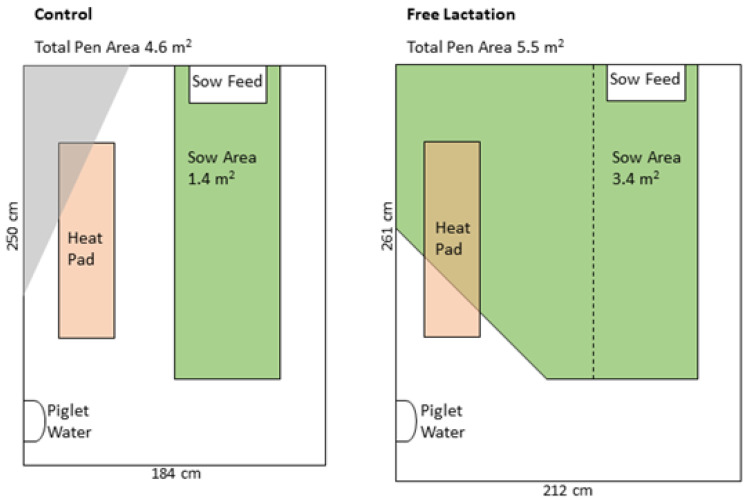
The Control and Free pen design. The area available to the sows in Control pens was 1.4 m^2^ and was the same in Free pens while the crate was closed, and 3.4 m^2^ when the crate was open. Water was available to sows ad libitum from a drinker located at the feed trough.

**Figure 2 animals-12-01762-f002:**
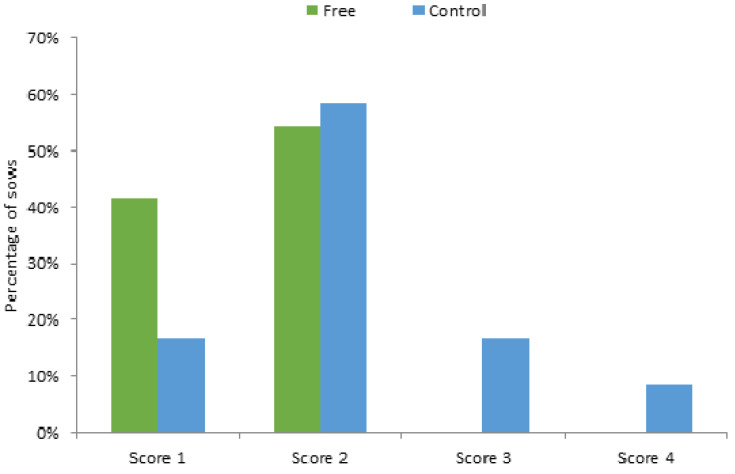
The percentage of sows that had locomotion scores 1, 2, 3 and 4 at weaning. No sows housed in the free lactation treatment had a score higher than 2 at weaning, while 17% of crated sows scored 3, and 8% scored 4 at weaning.

**Figure 3 animals-12-01762-f003:**
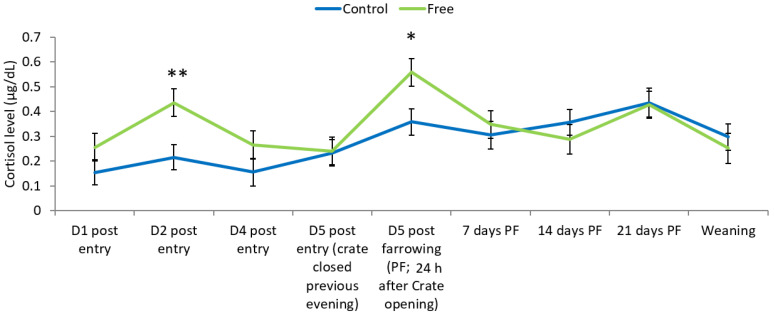
Salivary cortisol levels of sows in the Control (standard farrowing crate) or Free (sows enclosed in a crate from the onset of milk let-down until three days post farrowing) treatments throughout the experiment. Crates were opened three days post farrowing (PF). * indicates a tendency for a difference between treatments (*p* < 0.1 > 0.05), and ** indicates a significant difference (*p* < 0.05) on that recording day.

**Figure 4 animals-12-01762-f004:**
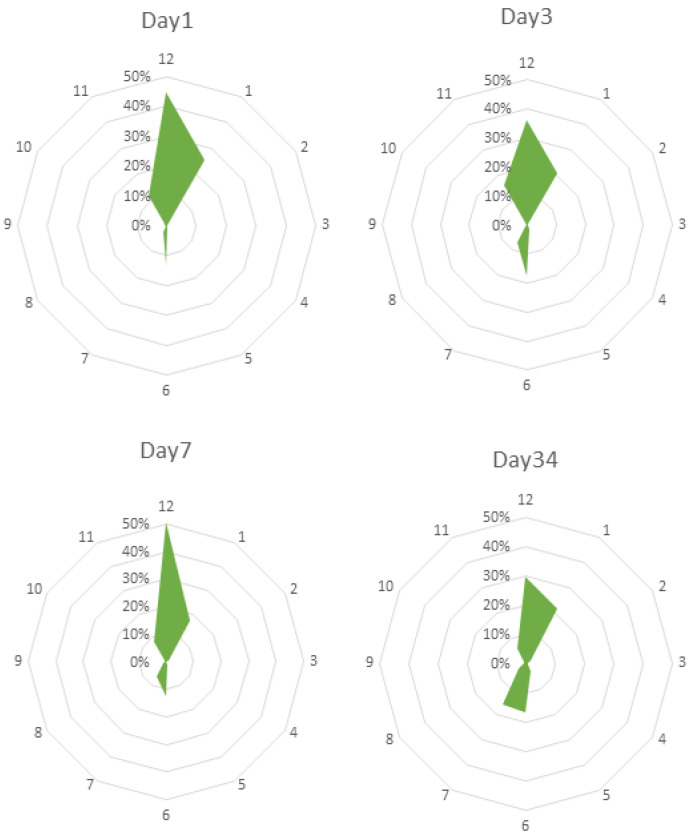
Sow orientation in Free pens on D1, D3, D7 and D34 (approximately day 25 of lactation) after entry. Position 12 represents the feed trough, and position 6 the rear wall of the pen. An increasing distance between the data point representing each position from the centre point of each graph indicates an increasing proportion of observations with the head oriented towards this position. The shaded area represents the proportion of observations for which sows were oriented in each direction.

**Table 1 animals-12-01762-t001:** Effect of management in a standard farrowing crate or a free lactation crate on live weight, back-fat depth and body condition loss during lactation.

	Control	Free	*p*-Value
Live weight (kg)			
Empty farrowing weight	232.39 ± 6.43	266.96 ± 6.56	0.50
Weaning	246.37 ± 3.74	241.44 ± 3.93	0.29
Back-fat depth (mm)			
Weaning	14.42 ± 0.42	14.40 ± 4.42	0.97
Lactation live weight loss (kg)			
Entry to weaning	−25.15 ± 4.18	−27.26 ± 4.27	0.69
Farrowing to weaning	−34.52 ± 1.94	−33.67 ± 1.98	0.72
Lactation back-fat loss (mm)			
Entry to weaning	2.21 ± 0.58	2.59 ± 0.59	0.60
Feed intake (kg)			
Total intake	170.8 ± 1.8	169.1 ± 1.9	0.44
Average daily intake	6.86 ± 0.07	6.77 ± 0.07	0.25

**Table 2 animals-12-01762-t002:** Hoof disorder scores for sows in the Control and Free treatments. Data are presented as medians and inter-quartile ranges.

	Control	Free	*p*-Value
Heel overgrowth and erosion	8 (6.25−9)	8 (6–10)	0.09
Heel–sole crack	8 (6–9.75)	9 (6−10)	0.10
White line damage	7 (5–8)	6 (5–8)	0.97
Dew claw length	6 (4–7)	5 (4–6)	0.12
Dew claw cracks	6 (3.25–8)	5 (3–7)	0.56
Vertical cracks	2 (1–4)	2 (1–3)	0.40
Horizontal cracks	1.5 (1–3)	2 (1–3)	0.40
Toe length	2 (2−2.75)	2 (2–2)	0.78

**Table 3 animals-12-01762-t003:** Tear stain scores for both the left and right eyes for sows in both treatments, at entry to the farrowing rooms and at weaning. Data are presented as medians and inter-quartile ranges.

	Control	Free	*p*-Value
Left Eye			
Entry	2 (1−3)	2 (1–3)	0.38
Weaning	2 (2–3)	2 (1–2)	0.05
Right Eye			
Entry	2 (1.5–3)	2 (1–3)	0.50
Weaning	2 (2–3)	2 (1–3)	0.29

## Data Availability

The data presented in this study are available on request from the corresponding author.

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
