# Peer review of "Freedom to Move: Free Lactation Pens Improve Sow Welfare"

_animals, 2022, doi:10.3390/ani12141762_

Round 1

Reviewer 1 Report

This is a well-described and easy-to-follow paper in which the sow welfare in free lactation crates was evaluated compared to sow in traditional farrowing crates. All the experimental procedures were correctly described, the statistical analysis was appropriate, and the discussion was perfectly developed.

However, a quiet comment about the cortisol measurement in saliva: the salivary cortisol assay kit (Salimetrics) should be validated for pig saliva. I am unsure if there is a study in which this assay has been previously validated in saliva from this specie. If there is, please, reference it. If not, the analytical validation should be performed by the authors. The following paper describes how an assay validation must be performed: FDA. Guidance for industry: bioanalytical method validation. Rocckville: US Department of Health and Human Service; 2001. http://www.fda.gov/ downloads/Drugs/GuidanceComplianceRegulatoryInformation/Guidances/ ucm070107.pdf. Accessed 16 Sept 2011.

Additionally, I miss the health status of the farm (vaccination, controlled for having free of what diseases, ..) or the breed of the selected sows.

Finally, the authors mentioned that the mortality and its causes were recorded daily. Where is that information in the result section? I miss details on crushing and the advantages or disadvantages between both systems in your results.

Reviewer 2 Report

General feedback:

The authors compare sow welfare and behavior in restricted vs. free farrowing systems. As they note such comparisons have been done already, but they include a four-day restriction to the free- farrowing system design during the time when piglet loss is considered highest, so this may offer a good compromise to reduce risks from crushing while increasing sow welfare.

The methods are not complete and currently several measures (e.g. contact with piglets) which are presented first in the results section should be introduced and described in the methods section.

The authors also mainly present cortisol as a stress indicator, without highlighting that cortisol is an indicator of arousal in response to negative as well as positive challenges, including for example exercise. This is touched on, but should be highlighted more as cortisol in this study could very well index eustress rather than distress/poor welfare.

 It is not always clear why the authors chose to analyze the behavioral and hormonal data as they did- e.g. why was each individual day included in models as a test predictor, rather than grouping days into for example “pre-“ and “post-day 4” when sows in the free farrowing system are able to move freely? It is also not clear why some measures were analyzed in a lateralized way rather than by combining measures- what is the importance of testing for laying on the left vs. right side, or for measuring tear staining in the left vs right eye? If there is not a good biological reason to split up these analyses, it may make more sense to combine them, and would reduce the risk of p-value inflation due to multiple tests.

Specific feedback:

Lines 127-131- it should be clarified how the locomotion score and back-fat thickness scores were determined.

Lines 156-158- How was hunger of piglets assessed? When was it deemed necessary to move them?

Line 248- the inter-assay CV is quite high for cortisol. I am wondering if this could have influenced the interpretation of the hormone results? For example, if all of the restricted-farrowing sows were measured on different plates from all of the free-farrowing sows, then the variation in cortisol between restricted and free-farrowing could be due in part to inter-plate noise. It would be good to mention in the methods how samples were organized on the plates. If this high CV is due to only 1 or 2 plates with strange values, it may be worth checking if the results stay the same when these outlier plates are removed.

Line 325-326- information about the calculation of sow posture should be added to the behavioral methods above.

Line 330-331- information about scoring of percentage of observations that sows were observed nursing should also be added to the behavioral methods

Lines 343-345- remove this text

Lines 358-363- it would be good to remind the reader that higher scores here mean more impaired locomotion, as this is a bit confusing. So rather than just stating the score, state what it means in terms of locomotion…

Results

Any reason for measuring tear stains in a lateralized way? Do you predict differences in the left vs right eye? Otherwise it would have made more sense to combine this analyses.

It seems that cortisol here may be linked to physical activity rather than to psychosocial stress- This should be emphasized more in the introduction on cortisol and in the results and discussion. Also, why use each day as a repeated measure rather than grouping the days into baseline and post-event when there were events of interest? Would be a more powerful way to test for changes in hormones…

469-477- There was no information in the methods about the measuring of piglet proximity- this should be added to the methods section.

Lines 561-563- cortisol levels vary for a variety of reasons including changes in diet and activity levels as well as in response to positive or negative challenges- therefore cortisol is a poor indicator of welfare on its own and it should not be assumed that high or low levels of cortisol necessarily indicate good or bad welfare states.

614-616- it is not only possible, but highly likely that cortisol is not a good proxy for welfare in this case and this should be emphasized throughout.

Round 2

Reviewer 1 Report

The authors have added all my previous concerns in the new version of the manuscript. Therefore, I accept the manuscript in its present form.